# Establishing the proportion of severe/ moderately severe vs mild cases of progressive disseminated histoplasmosis in patients with HIV

**Mathieu Nacher**[1,2]*, Antoine Adenis[1,2], Romain Blaizot[3,4], Philippe Abboud[5], Paul Le Turnier[5], Ugo Françoise[1], Aude Lucarelli[6], Magalie Demar[4,7], Félix Djossou[5], Loïc Epelboin[5], Pierre Couppié[2,3]

**1** CIC INSERM 1424, Centre Hospitalier de Cayenne, Cayenne, French Guiana, **2** Département Formation Recherche, Université de Guyane, Cayenne, French Guiana, **3** Service de Dermatologie, Centre Hospitalier de Cayenne, Cayenne, French Guiana, **4** Unité mixte de recherche Tropical Biome and Immunopathology, Université de Guyane, Cayenne, French Guiana, **5** Service des maladies infectieuses et tropicales, Centre Hospitalier de Cayenne, Cayenne, French Guiana, **6** Coordination Régionale de la lutte contre le VIH et les infections sexuellement transmissibles Guyane, Centre Hospitalier de Cayenne, Cayenne, French Guiana, **7** Laboratoire de Parasitologie-Mycologie, Centre Hospitalier de Cayenne, Cayenne, French Guiana

* mathieu.nacher@ch-cayenne.fr

**Data Availability Statement:** The data may be made available for reasonable requests after obtaining the authorization from the commission

## Abstract

### Background

Progressive disseminated histoplasmosis remains a major but neglected cause of death among patients with advanced HIV. Recently, aiming to reduce avoidable deaths, the Pan American Health Organization issued the first diagnosis and treatment guidelines for HIV-associated histoplasmosis. But what proportion of progressive disseminated histoplasmosis in HIV-infected patients is severe is currently not known. Because this proportion influences treatment needs, we aimed to estimate this in a cohort of 416 patients in French Guiana.

### Methods

We used the definition in the recent PAHO/WHO guidelines for severity. We used regression modelling to predict the impact of CD4 count on the proportion of severe cases. In a territory where treatment cost is not a limiting factor and where histoplasmosis is well known, we assumed that clinicians' initial treatment reflected their perception about the severity of the case and therefore, the needs for different treatments.

### Results

Using these definitions, since the beginning, there were 274 (65.9%) severe/moderately severe cases and 142 (34.1%) mild cases. In practice 186 cases were treated with deoxycholate or liposomal amphotericin B (44.7%) and 230 (55.3%) cases treated with itraconazole as first line treatment. The Kappa concordance measure between the guideline definition and the actual treatment given was 0.22. There was a 9% risk difference for death

informatique et libertés, 3 Pl. de Fontenoy, 75007 Paris

**Funding:** The author(s) received no specific funding for this work.

**Competing interests:** The authors have declared that no competing interests exist.

within 30 days of antifungal treatment initiation between severe/moderately severe and mild cases. Over threequarters (77%) of early deaths were attributed to severe/moderately severe cases.

## Conclusions

This is the only rigorous estimate of the proportion of severe/moderately severe cases of progressive disseminated histoplasmosis in symptomatic HIV patients on the largest published cohort. These numbers may help defend budget needs for rapid diagnostic tests and liposomal amphotericin B.

### Author summary

What proportion of progressive disseminated histoplasmosis in HIV-infected patients is severe is currently not known. Because this proportion influences treatment needs, we aimed to estimate this in a cohort of 416 patients in French Guiana. We used the definition in the recent PAHO/WHO guidelines for severity. In a territory where treatment cost is not a limiting factor and where histoplasmosis awareness is high, we assumed that clinicians' initial treatment reflected their perception of severity and therefore, the needs for different treatments. Using these definitions, since the beginning, there were 274 (65.9%) severe/moderately severe cases and 142 (34.1%) mild cases. In practice 186 cases were treated with deoxycholate or liposomal amphotericin B (44.7%) and 230 (55.3%) cases treated with itraconazole as first line treatment. The Kappa concordance measure between guidelines and treatment was 0.22. There was a 9% risk difference for death within 30 days of antifungal treatment initiation between severe/moderately severe and mild cases. Over threequarters (77%) of early deaths were attributed to severe/moderately severe cases. This is the only rigorous estimate of the proportion of severe/moderately severe cases of progressive disseminated histoplasmosis in symptomatic HIV patients on the largest published cohort. These numbers may help defend budget needs for rapid diagnostic tests and liposomal amphotericin B.

## Introduction

Progressive disseminated histoplasmosis has been on the list of AIDS-defining opportunistic infections for over 3 decades but it is still widely underdiagnosed; this has resulted–and still results in—a great burden of avoidable deaths in Latin America and presumably on other continents [1]. In a nutshell, there is a diagnostic problem, which for individual patients leads to the absence or delays in diagnosis resulting in significant risk of death; it also leads to a lack of data, knowledge, and awareness among physicians. This has mobilized researchers, clinicians, and public health specialist around the Manaus declaration which set the target of 100% of hospitals in Latin America having the diagnostic capacity and the recommended antifungal drugs by 2025 to promptly diagnose and treat histoplasmosis and, ultimately, reduce mortality among patients with advanced HIV disease [2]. After establishing the first progressive disseminated histoplasmosis diagnosis and treatment guidelines for patients with HIV, WHO/PAHO are now pushing, despite the COVID-19 epidemic, for the rollout of tests and drugs and thus discussing with the industry and country ministries to reach the target [3]. However, in a

context where many countries have limited resources, who should benefit from histoplasmosis antigen detection tests is not yet clearly defined. Although most studies actually concern patients with patent disease, it is often believed that the natural history of progressive disseminated histoplasmosis unfolds progressively and that in the early stages many patients have few or no symptoms until the fungal burden reaches symptomatic levels, a scenario that follows what is known for cryptococcal meningitis [4]. The reason why progressive disseminated cases in most studies are severe or moderately severe is that a substantial proportion of persons with HIV are still diagnosed at the advanced stage and that, in most of Latin America, antigen detection tests have been missing. In fact, we do not even know what proportion of progressive disseminated histoplasmosis in HIV-infected patients are severe and what proportion are moderately severe or mild, a proportion that influences treatment needs for an expensive drug —liposomal amphotericin B. Indeed, as the WHO is attempting to model the needs for tests and drugs to achieve the aims of the Manaus declaration, these numbers are crucial to refine estimated budgeting to realistically cover diagnostic and treatment needs. Progressive disseminated histoplasmosis is the most frequent opportunistic infection in patients with advanced HIV [5,6]. In this context, we have compiled a unique hospital cohort of histoplasmosis cases since the 1990s; we used this data to estimate the proportion of severe/moderately severe cases and of mild cases according to the recent WHO/PAHO definition. Since the guidelines recommend the use of liposomal amphotericin B as a first line treatment in severe and moderately severe cases, we looked at concordance between the severity according to WHO/PAHO guidelines and physicians' first line antifungal treatment choice. We also estimated 1-month case fatality in the severe/moderately severe versus the mild histoplasmosis group.

## Materials and methods

### Ethics statement

This project has received approval by: the Comité Consultatif pour le Traitement de l'Information pour la Recherche en Santé (CCTIRS) (no. 10.175bis, 6 October 2010); the French National Institute of Health and Medical Research Institutional Review Board (CEEI INSERM) (IRB0000388, FWA00005831 18 May 2010); and the Commission Nationale Informatique et Libertés (CNIL) (no. JZU0061856X, 16 July 2010).

### Study design

We retrospectively analyzed HIV-infected patients with confirmed disseminated histoplasmosis between January 1st 1981 and 31st December 2020 in the 3 hospitals of French Guiana; Cayenne, Kourou, and Saint Laurent du Maroni.

### Study population

The inclusion criteria were: confirmed HIV infection; first proven episode of histoplasmosis (European Organization for Research and Treatment of Cancer/Mycoses Study Group (EORTC/MSG) criteria); and age >18 years [7].

### Study conduct

The histoplasmosis database includes incident cases of HIV-associated histoplasmosis (described in more detail in reference [8]). Diagnosis of histoplasmosis was performed using direct examination and culture of samples obtained by aspiration or biopsy. Diagnosis was also obtained by cytopathology and histopathology using Gomori–Grocott staining; molecular diagnosis was not routinely available and antigen detection was not available.

### Defining severe

The retrospective database did not have a variable labeling patient as "severe", "moderately severe" or "mild". Using the criteria from the WHO/PAHO histoplasmosis guideline we generate a variable that classified patients as severe or moderately severe in "the presence of at least one sign or symptom involving vital organs: respiratory or circulatory failure, neurological signs, renal failure, coagulation anomalies and a general alteration of the WHO performance status greater than 2" [9].

To determine whether this definition was concordant with what clinicians actually prescribe (clinicians in French Guiana are well aware of the high incidence of histoplasmosis, its associated risk of death, and the need for aggressive diagnostic efforts and presumptive treatment) we calculated a kappa coefficient to quantify concordance. This allowed us to classify patients in our cohort in 2 groups: "severe or moderately severe" and "mild" to look at the risk of dying within 1 month after initiating antifungal treatment–which is a better reflection of case fatality than all mortality as patients with advanced HIV may also die from other causes after histoplasmosis is treated.

### Statistical analysis

The crude percentage of Severe/moderately severe histoplasmosis and mild histoplasmosis were computed; Given the importance of CD4 counts we modeled the risk of being severe or moderately severe using logistic regression followed by marginal effects predictions by increments of 10 CD4 cells and plotted the curve and 95% confidence intervals until 350 CD4 per mm3. A variable representing the difference between the year of histoplasmosis treatment relative to HIV diagnosis was generated to study the distribution of histoplasmosis cases relative to HIV diagnosis. Case fatality rates at 1 month and the risk difference were computed for severe/moderately severe and mild patients after liposomal amphotericin B was available (1999). Concordance between the PAHO definition and the physician's first line treatment was quantified with Kappa's coefficient.

Statistical significance was set at $p < 0.05$. STATA 16 software (College Station, TX, USA) was used for statistical analysis.

## Results

Overall, since the beginning, there were 274 (65.9%) severe cases and 142 (34.1%) non severe cases. When taking into account the CD4 counts, the proportion of severe ranged from between 40% among those with the highest CD4 count and 70% among those with the lowest CD4 count (Fig 1). Table 1 shows there were no striking demographic differences between severe/moderately severe and mild HIV-associated disseminated histoplasmosis.

### When does histoplasmosis occur relative to HIV diagnosis?

Fig 2 shows that over half of patients were diagnosed for histoplasmosis around the same time period that HIV was diagnosed. Most patients with histoplasmosis were not on antiretrovirals 354/416 (85%) at the time of diagnosis. Among those on antiretroviral treatment there was no significant difference in the proportion of severe cases (59.7% vs 66.9%, P = 0.26).

### Early death

Between 1999 and 2020, when liposomal amphotericin B was available, there were 28 deaths within 30 days after initiating antifungal drugs.

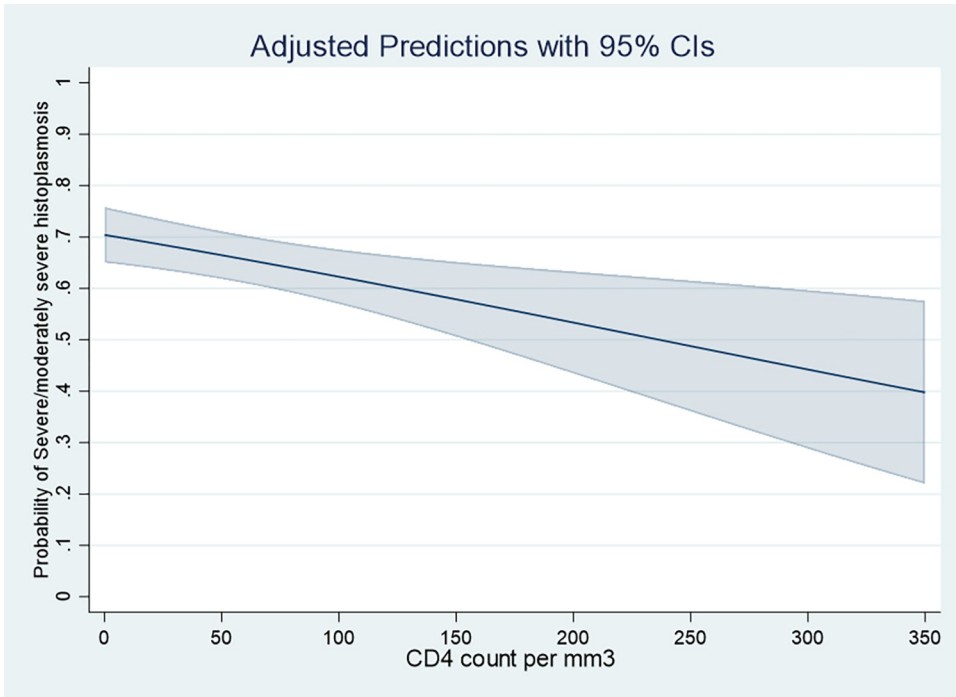

**Fig 1. Probability of severe progressive disseminated histoplasmosis by CD4 count.**

Table 2 shows the incidence of death at one month between severe/moderately severe and mild progressive disseminated histoplasmosis among HIV positive patients. Death was more frequent in the severe/moderately severe group and the 0.09 risk difference suggested that for every 11 patients that were mild instead of severe/moderately severe, this would avoid one death within 30 days. The 77% attributable fraction of severity in the overall progressive disseminated histoplasmosis population suggests that if there were no more severe presentations over three quarters of deaths at 1 month would be suppressed.

## Physicians first line treatment choice

For physicians, 186 cases were treated with deoxycholate or liposomal amphotericin B (44.7%) and 230 (55.3%) cases were treated with itraconazole as first line treatment. Among those 186

**Table 1. Sociodemographic data in severe/moderately severe vs mild HIV-associated disseminated histoplasmosis.**

|  | Severe/moderately severe disseminated histoplasmosis* | Mild disseminated histoplamosis* | P |
|---|---|---|---|
| **Mean age (SD) in years** | 40.5 (9.9) | 40.8 (9.9) | 0.77 |
| **Sex N(%)** |  |  | 0.97 |
| M | 178 (65) | 92 (64.8) |  |
| F | 96 (35) | 50 (35.2) |  |
| **Nationality N(%)** |  |  | 0.13 |
| France | 75 (28) | 38 (27.3) |  |
| Brazil | 47 (17.5) | 22 (15.8) |  |
| Haiti | 51 (19) | 35 (25.2) |  |
| Suriname | 62 (23.1) | 21 (15.1) |  |
| Guyana | 15 (5.6) | 10 (7.2) |  |
| Other | 13 (4.8) | 5 (3.6) |  |

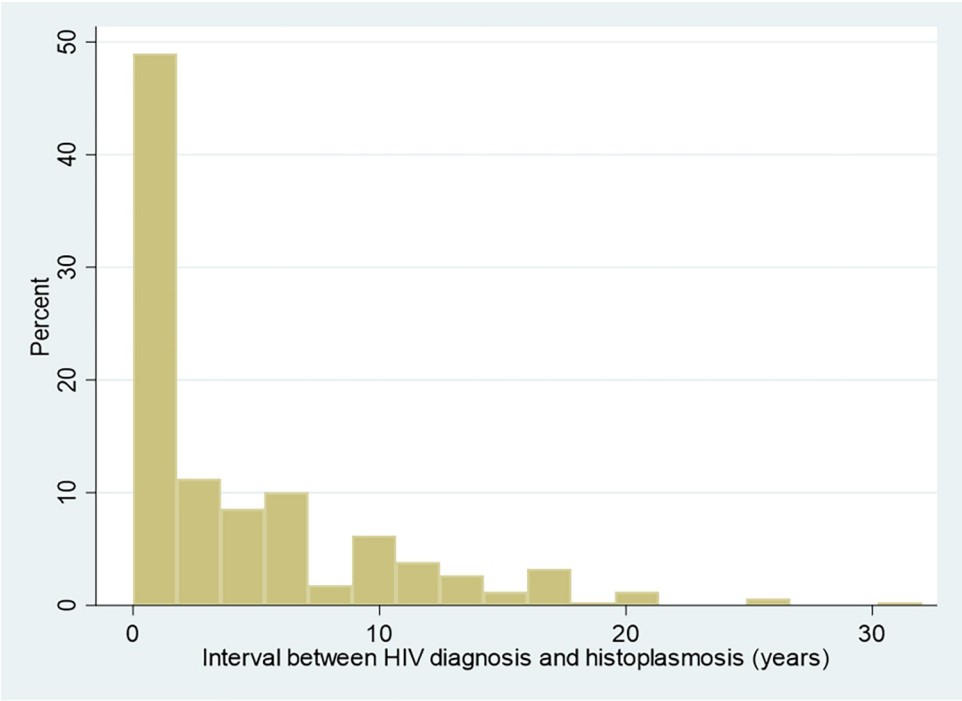

**Fig 2. Interval between HIV diagnosis and progressive disseminated histoplasmosis.**

patients receiving deoxycholate or liposomal amphotericin B, 93 patients received a double therapy with itraconazole as part of the first line regimen. Table 3 shows death within one month of antifungal therapy by physician choice.

The concordance between actual first line antifungal treatment and what the PAHO/WHO recommendations would have prescribed showed a 59.6% agreement and a "fair" Kappa at 0.22, (P<0.0001).

Fig 3 shows the Venn diagram representing the concordance of first line treatment according to severity according to PAHO/WHO guidelines. Among patients with severe/moderately severe histoplasmosis, 101 were treated with itraconazole only, and 6 (5.9%) died within 1

**Table 2. Risk of death at 1 month for severe/moderately severe or mild progressive disseminated histoplasmosis according to the PAHO definition and for actually given first line treatment.**

|  | Severe/moderately severe disseminated histoplasmosis* | Mild disseminated histoplasmosis* |
|---|:---:|:---:|
| **Dead within 1 month of antifungal treatment initiation** | 26 | 2 |
| **Alive 1 month after antifungal treatment initiation** | 202 | 103 |
| **Total** | 228 | 105 |
| **Risk** | 0.11 | 0.02 |
| **Risk difference** | 0.09<br>(95%CI = 0.04–0.14) |  |
| **Risk ratio** | 6 (95%CI = 1.4–24.7), P = 0.003 |  |
| **Attributable fraction among the severe** | 0.83 |  |
| **Attributable fraction in all disseminated histoplasmosis cases** | 0.77 |  |

*2020 WHO/PAHO definition: presence of at least one sign or symptom involving vital organs: respiratory or circulatory failure, neurological signs, renal failure, coagulation anomalies and a general alteration of the WHO performance status greater than 2

**Table 3. Death within one month by first line antifungal treatment choice.**

|  | First line treatment Liposomal amphotericin B | First line treatment Itraconazole |
|---|---|---|
| **Dead within 1 month of antifungal treatment initiation** | 20 | 8 |
| **Alive 1 month after antifungal treatment initiation** | 140 | 165 |

month of antifungal treatment initiation, while the guidelines would have recommended liposomal amphotericin B.

## Discussion

Here we show, in the largest cohort of HIV-associated progressive disseminated histoplasmosis, that among symptomatic patients, 65.9% of cases were severe or moderately severe. We show that this proportion increases as CD4 counts decline. WHO now recommends prompt antiretroviral treatment of HIV patients, sometimes the same day as the day of the HIV diagnosis [10]. As an opportunistic infection, progressive disseminated histoplasmosis is linked to the level of immunosuppression, which is reversed by antiretroviral treatment and virological suppression. Indeed, after adjusting for multiple variables, antiretroviral treatment was associated with an 80% reduction of incidence of progressive disseminated histoplasmosis, and among those with histoplasmosis mortality was also reduced by 80% among patients with histoplasmosis on antiretroviral treatment [11]. Here, 85.1% of patients with progressive disseminated histoplasmosis were not on antiretroviral treatment when diagnosed, and over half of histoplasmosis diagnoses were given within the year of the diagnosis. This implies that

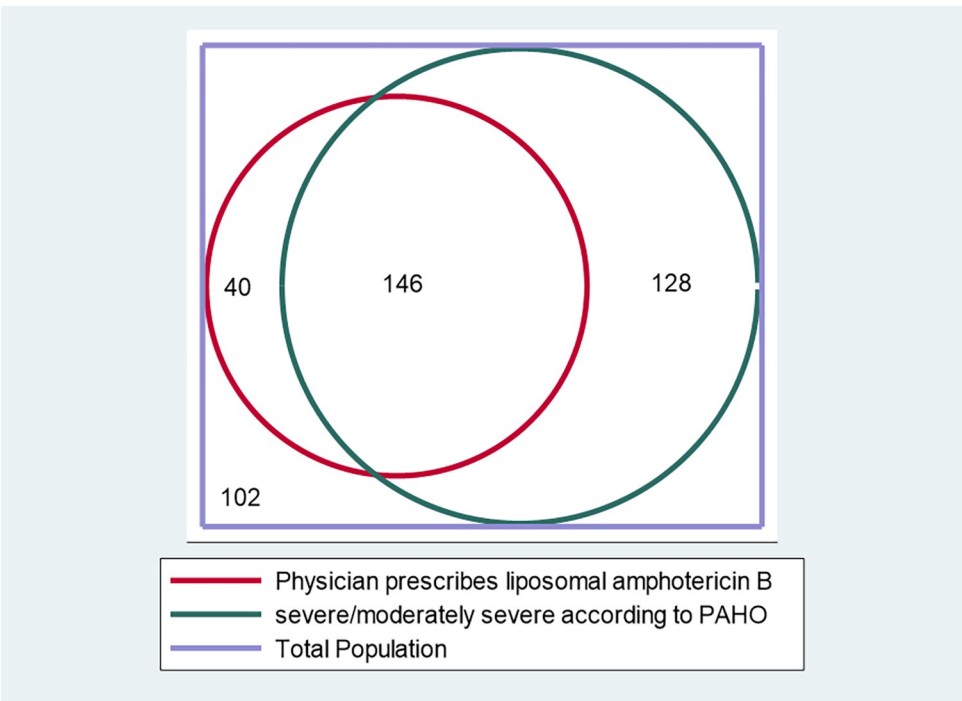

**Fig 3. Venn diagram representing the concordance between the PAHO/WHO guideline definition of severe/moderately severe and first line treatment.**

although there have been great advances in treating HIV, the bulk of the patients with advanced disease and its complications comes from the hidden reservoir of persons unaware of their HIV infection and that HIV testing efforts are also key to reduce the burden of histoplasmosis. In French Guiana, HIV testing is sustained with 203 tests per 1000 inhabitants per year, but still around a third of patients are diagnosed at the advanced disease stage; this proportion is often higher in many parts of Latin America with lower testing activities [12,13]. Thus, cascades of care involving HIV testing activity, patient orientation and treatment, and retention in care are key elements that may vary between countries and will influence the number of progressive disseminated histoplasmosis cases and the proportion of severe/moderately severe and thus initial treatment therapeutic needs. In addition to the depth of immune-suppression, the dissemination of *H. capsulatum* is also favored by lack of awareness and fungal diagnostics which further delay treatment and lead to greater fungal burdens and severity. Both late HIV diagnosis and late fungal diagnosis are both related to health care resources and thus presumably shift the proportion of histoplasmosis cases towards severe presentations.

Here we show that among HIV-infected patients in French Guiana, where availability and cost of treatment is not a limiting factor, 44.7% of progressive disseminated histoplasmosis cases were initially treated with an injectable amphotericin B formulation, mostly liposomal. The Kappa concordance measure was only fair [14] as over 30% of severe/moderately severe were treated by itraconazole. This shows that, if we refer to the recent PAHO/WHO guidelines, there was a substantial gap between recommendations and practice. Hence, although liposomal amphotericin B is easily available, it seemed under-prescribed, with possible detrimental consequences for some patients. In most of Latin America, it is not available. Estimates of disseminated histoplasmosis in HIV patients in Latin America in [15] suggest there are between 6710 (if 30% of incident cases are symptomatic) and 15647 (if 70% of incident cases are symptomatic) incident symptomatic cases. Here, with over 65% of cases being severe/moderately severe, this means that between 4361 and 10170 patients should receive a course of liposomal amphotericin B. In French Guiana, a 50mg vial is 120 Euros, if we assume 4 vials per day for 10 days this represents, 4800 euros for the course. Therefore, at France's prices, the total cost would range between 22,252,800 and 48,818,640 Euros. In low- and middle-income countries the costs have been reduced to 15.4 Euros for cryptococcal meningitis [16]. Such a price for disseminated histoplasmosis would divide the cost nearly 8-fold. However, these coarse calculations are not proper cost-effectiveness studies; they are crude attempts to emphasize the economic correlates and by no means should be taken as a definitive figure.

Among the limitations, although we used the PAHO/WHO definition, the boundary between moderately severe and mild cases is a fuzzy one and, away from the extremes, there seems to be no clear way to distinguish the two. Furthermore, this is a retrospective study covering a wide time-span during which many gradual changes may have occurred, such as improved diagnostic capacity, physician experience, and diagnostic and treatment decisions; the synthesis of this in crude measures fails to capture these secular trends. Another potential limitation is that the use of the risk difference to calculate an equivalent of the "number needed to treat" and the attributable fractions within the populations may be criticized because, given the number of confounders, causal links are not established. However, we think that it may also be an interesting parameter of future models if we consider that screening patients with advanced HIV will allow an earlier diagnosis and hence a lower case-fatality. The 77% attributable fraction in the population shows that over three quarters of early mortality could be avoided if severe/moderately severe presentations were replaced by mild cases—through earlier diagnosis. Furthermore, ultimately screening of all patients under 200 CD4 per mm3 (or 350 as suggested by studies in Guatemala [17]) would even be expected to further reduce early

death by avoiding severe/moderately severe and even mild presentations, which we see here are also associated with early death.

Overall, given the gaping absence of data on the topic, the present results offer, as a start, some parameters that could (in conjunction with country incidence rates, proportions with advanced HIV. . .) help estimate treatment needs and expected benefits of improvement of diagnosis and care of disseminated histoplasmosis in HIV-infected patients. The needs for affordable liposomal amphotericin B and rapid diagnostic tests are considerable and no efforts should be spared to achieve the goals of the Manaus declaration.

## Author Contributions

**Conceptualization:** Mathieu Nacher.

**Data curation:** Antoine Adenis, Ugo Françoise.

**Formal analysis:** Mathieu Nacher.

**Investigation:** Romain Blaizot, Philippe Abboud, Paul Le Turnier, Aude Lucarelli, Magalie Demar, Félix Djossou, Loïc Epelboin, Pierre Couppié.

**Writing – original draft:** Mathieu Nacher.

**Writing – review & editing:** Mathieu Nacher, Antoine Adenis, Romain Blaizot, Philippe Abboud, Paul Le Turnier, Ugo Françoise, Aude Lucarelli, Magalie Demar, Félix Djossou, Loïc Epelboin, Pierre Couppié.

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
