## [Decision Letter · Decision Letter 0]

26 Sep 2022

Dear Dr. Nacher,

Thank you very much for submitting your manuscript "Establishing the proportion of severe/moderately severe vs mild cases of progressive disseminated histoplasmosis in patients with HIV" for consideration at PLOS Neglected Tropical Diseases. As with all papers reviewed by the journal, your manuscript was reviewed by members of the editorial board and by several independent reviewers. The reviewers appreciated the attention to an important topic. Based on the reviews, we are likely to accept this manuscript for publication, providing that you modify the manuscript according to the review recommendations. 

Sincerely,

Angel Gonzalez, Ph.D.

Academic Editor

Marcio Rodrigues

Section Editor

Reviewer's Responses to Questions

**Key Review Criteria Required for Acceptance?**

**Methods**

-Are the objectives of the study clearly articulated with a clear testable hypothesis stated?

-Is the study design appropriate to address the stated objectives?

-Is the population clearly described and appropriate for the hypothesis being tested?

-Is the sample size sufficient to ensure adequate power to address the hypothesis being tested?

-Were correct statistical analysis used to support conclusions?

-Are there concerns about ethical or regulatory requirements being met?

Reviewer #1: The study design is appropriate.

The objectives are well displayed and articulated with methods.

The sample size is substantial. All subjects are from one tropical territory.

More demographic data is advisable.

The authors used simple and efficient statistical analysis.

There is no concern about ethical or regulatory requirements.

Reviewer #2: The manuscript "Establishing the proportion of severe/moderately severe vs mild cases of progressive disseminated histoplasmosis in patients with HIV" is very interesting and clearly shows the difficulty in establishing the thin line between severe and moderately severe cases versus mild cases when the physician caring of any particular patient receives the patient, diagnoses HIV infection as well as histoplasmosis infection at the first contact.

This huge clinical difficulty is described in this manuscript and the methods used for this evaluation are adequated. 

Therefore the study design addressed the objectives to prove the initial hypothesis tested, and used adequated analysis methods to achieve the goal.

As this study is retrospective, there are no concerns about the ethical requirements.

Due to the fact that histoplasmosis infection is not so common, even among HIV+ patients, the range of time (40 years) led to the possibility of modifications in the proposed therapy for the fungal disease, as well as HIV infection. 

This is a very difficult confounder to evaluate. In order to exclude this confounder the authors used the whole number of cases through all the time to study the distribution of histoplasmosis cases relative to HIV diagnosis, and calculated the death risk difference at one month of diagnosis only for the patients diagnosed after 1999, when liposomal amphotericin B was available.

Moreover, the therapy of HIV infection evolved significantly in this time, allowing nowadays that HIV+ patients treated and with a good adherence to the therapy are protected from the acquisition of severe infections by opportunistic pathogens such as Histoplasma spp.

**Results**

-Does the analysis presented match the analysis plan?

-Are the results clearly and completely presented?

-Are the figures (Tables, Images) of sufficient quality for clarity?

Reviewer #1: The results are well exposed.

Tables 1 and 2 could be more reader-friendly if they had horizontal stripes individualizing the lines.

Images are well conceived.

Reviewer #2: The results are clearly presented and described in the text, and the figures are adequately clear to the readers helping them to understand quickly the data presented in the manuscript.

**Conclusions**

-Are the conclusions supported by the data presented?

-Are the limitations of analysis clearly described?

-Do the authors discuss how these data can be helpful to advance our understanding of the topic under study?

-Is public health relevance addressed?

Reviewer #1: Data presented supports conclusions from an appealing point of view and with a simple and nonordinary focus. 

Discussion competently addresses the results for public health, sometimes with some "not so suitable" extrapolations, like cost calculations, which were not researched in this study.

The description of study limitations should be more precise. The authors could reserve more words for this.

Reviewer #2: The conclusions are supported by the data, and the discussion of the results are adequate and helpful to explain to the reader the tough decisions presented to the clinical physician caring the HIV patients presenting with histoplasmosis.

**Editorial and Data Presentation Modifications?**

Reviewer #1: The authors inform that data will be avaiable for reasonable requests. Editors could consider asking about wich conditions are they. Maybe PLoS could solve them, or at least provide enhanced guidance to the scientists.

Tables 1 and 2 could be more reader-friendly if they had horizontal stripes individualizing the lines.

Reviewer #2: No, the text is very well written and clear, and the presentation of data adequate to help the reader in the decision of therapy when he/she is presented to similar cases.

**Summary and General Comments**

Reviewer #1: The topic under study received a concrete contribution from this paper. Histoplasmosis in the setting of HIV/AIDS needs more knowledge about it, particularly in the tropics.

Statistical analyses are well explained and simplified to the reader, which is an excellent marker. The idea of showing attributable fractions is a simple way to clarify data to the reader.

Minor reviews are addressed in anterior fields:

- Presentation of more detailed demographic data;

- Minimize atrapolations about cost;

- A more careful discussion about study limitations.

I hope I have helped and salute the authors for the well-qualified study.

Reviewer #2: Summarizing the review, in my point of view this manuscript is clinically very interesting and adequately executed, allowing it to be published at PLOS NTD.

PLOS authors have the option to publish the peer review history of their article (what does this mean?). If published, this will include your full peer review and any attached files.

Reviewer #1: Yes: Alberto dos Santos de Lemos

Reviewer #2: No

Figure Files:

Data Requirements:

Reproducibility:

References

---

## [Editor Report · Decision Letter 1]

29 Sep 2022

Dear Pr. Nacher,

We are pleased to inform you that your manuscript 'Establishing the proportion of severe/moderately severe vs mild cases of progressive disseminated histoplasmosis in patients with HIV' has been provisionally accepted for publication in PLOS Neglected Tropical Diseases.

Best regards,

Marcio L Rodrigues

Section Editor

Marcio Rodrigues

Section Editor

---

## [Editor Report · Acceptance letter]

31 Oct 2022

Dear Pr. Nacher,

We are delighted to inform you that your manuscript, "Establishing the proportion of severe/moderately severe vs mild cases of progressive disseminated histoplasmosis in patients with HIV," has been formally accepted for publication in PLOS Neglected Tropical Diseases.

Best regards,

Shaden Kamhawi

co-Editor-in-Chief

Paul Brindley

co-Editor-in-Chief
